# Timber Structures and Prefabricated Concrete Composite Blocks as a Novel Development in Vertical Gardening

**Tõnis Teppand [1], Olesja Escuer [2,3], Ergo Rikmann [4], Jüri Liiv [1,4] and Merrit Shanskiy [2,\*]**

1   Institute of Forestry and Rural Engineering, Estonian University of Life Sciences, Fr. R. Kreutzwaldi 5, 51006 Tartu, Estonia
2   Institute of Agricultural and Environmental Sciences, Estonian University of Life Sciences, Fr. R. Kreutzwaldi 5, 51006 Tartu, Estonia
3   Botanical Garden, University of Tartu, Lai 38, 51005 Tartu, Estonia
4   Institute of Chemistry, Faculty of Science and Technology, University of Tartu, Ravila 14a, 50411 Tartu, Estonia
\*   Correspondence: merrit.shanskiy@emu.ee

**Abstract:** A modern, environmentally friendly urban lifestyle requires paying attention to landscaping and green areas. The scarcity of free land in cities and the high price of land require the combination of greenery with buildings—both vertically and horizontally. The developed green technology for construction brings together computer numerical control (CNC) processing of supporting structures and prefabricated solid planting blocks made of concrete composite. The timber structures are fixed together using traditional carpentry joints. The details, which will be manufactured in the factory using CNC processing at a controlled temperature and humidity corresponding to indoor conditions, can be easily assembled on the construction site. The high bending strength but good elasticity and connections of carpentry joints endow the structure with good properties in a non-controllable environment. By combining CNC-processed wooden structures with concrete technology as substrate composites, labor-intensive manual work in landscaping and gardening will be reduced in the future. The novel material-hardening substrate composite material uses only the residues as the raw materials.

**Keywords:** concrete composite substrate; CNC machining; modular assembly; carpentry joints; vertical gardening

## 1. Introduction

Vertical landscape refers to vegetation that grows directly on the facade of a building, on specially designed structures or on internal walls. They can consist of vegetation mats, modules, or trellises that will be installed on building walls or attached to frames. Considered to be part of the sustainable urban concept, vertical landscapes are aesthetically pleasing, allowing the addition of greenery to the cityscape by taking advantage of vertical surfaces: they can be used in parks and public facilities and built-up commercial, community, educational and healthcare buildings. They bring additional color to the urban environment [1].

Vertical gardening in the urban landscape has many ecological benefits, in addition to social and architectural value [2]. A study carried out in Quito, Ecuador, showed that vertical gardens contributed to significant improvement in outdoor air quality regarding the concentrations of particles, $O_3$, and $NO_2$ [3]. Green roofs and vertical gardens contribute to the sustainable management of urban runoff and mitigation of the urban heat island effect. Moreover, they allow the abatement of noise, reduction in air pollution (including fine particles), improved overall ecological preservation due to microhabitats created for flora and fauna [4], visual valorization of unattractive large "empty" facades and walls, and combating graffiti [5].

Indoor farming is a rapidly expanding business in Europe. This growth is driven by increasing consumer demand for locally grown, healthy and fresh food. An important prerequisite is also the cheapening of LED lighting technologies. Economic recessions may give a new impetus to indoor farming, as alternative uses are sought for vacant industrial or office buildings. This phenomenon was observed during and after the 2008–2009 financial crisis [6].

In addition to aesthetic aspects, indoor farming and indoor design elements with vegetation can significantly improve indoor air quality [7] and reduce the sick building syndrome [8]. The time people spend indoors reaches up to 80% in modern urban areas. Therefore, indoor air quality control is even more important from the point of view of public health than atmospheric air pollution control. The pollution level of indoor air often exceeds the outdoor air pollution by 2–5 times, and sometimes in poorly ventilated rooms, it reaches up to 100 times [9]. Indoor vegetation has been found to reduce the count of microorganisms up to five times in the air [10]. Vertical farming, where plant photosynthesis relies on daylight, allows for a quarter to a third reduction in indoor $CO_2$ concentration in a typical 30 m$^2$ office with 1–3 occupants. This, in turn, ensures significant savings on ventilation costs. When rapidly growing photophilic vegetables were vertically farmed, the $CO_2$ drawdown was markedly higher in comparison to sciophilous landscaping plants with a lower net photosynthetic rate. The difference was up to nine times due to differences in photosynthesis rates [7]. The phytoremediation of indoor air is also an effective natural method for a number of other pollutants. For example, potted plants and green walls absorb volatile organic compounds from room air [9]. All these advantages of vertical gardening contribute to a more inhabitable environment, as well as to the reduction in energy consumption in buildings and thus to the goals of the Green Revolution. This is important, among other things, in view of the increase in the price of energy carriers and electricity. Architectural engineering models have been developed for the vertical landscaping of outer surfaces/facades and roofs in semi-arid climates, based on multivariate landscape–ecological analysis. First, the problems that the proposed design intended to solve will be defined, and the activities necessary for this will be planned. This forms a basis on which an "objective tree" is constructed. Then, the criteria for an optimized solution, taking into consideration various factors, including but not limited to, climate-related, plant-related, architectural, and safety factors were used to compare the alternatives pairwise by creating a matrix of binary comparisons. Based on this matrix, vectors of local priorities were calculated using a mathematical procedure described by N. Ivanova et al. [5]. Local priority vectors were then used to calculate the global priority vectors for each element. As a result, a three-level hierarchical model was obtained where each level, in turn, divided into sublevels. The model creates a dynamic hierarchy and each level, in turn, divided into sublevels. Model-calculated integrated priorities allowed the qualitative ranking of alternative design options [5].

Some of the most common technical solutions for vertical garden systems, both indoor and outdoor, are continuous elements based on felt or geotextile, and discontinuous panel systems, where plants grow directly out of a vertically placed element, and container systems, where plants grow in containers installed on the facade or parallel to the façade. A number of engineering solutions and mathematical models has been developed to integrate green walls with building technical systems, in particular with indoor climate control systems [11] and other "smart house" features [12]. Green walls have been modeled significantly to improve the buildings' energy performance and thermo-hygrometric comfort of their occupants in the Mediterranean climate [13]. Indoor greenery and vertical farming are considered key elements in achieving indoor environmental quality in high-rise buildings. In addition to decreasing the energy demand on cooling and ventilation, this allows to improve indoor acoustic conditions, produce fresh food for in situ consumption, contribute to sustainable stormwater management, and, in combination with in situ energy generation, improve the overall sustainability of skyscrapers [14].

However, the existing engineering solutions described in the literature typically rely on the use of substantial amounts of plastic and metal parts [11] and details. For the sake of sustainability, it is important that vertical greening uses as many green, carbon-neutral, yet durable and affordable materials as possible. It is also important that these materials are aesthetically pleasing and match the interior design of the rooms and architecture of the buildings and urban landscapes. In addition to structural materials, plant-growing substrates and nutrients must be from renewable sources.

This paper proposes a new conceptual approach to modern methods of designing and maintaining vertical gardens that addresses all of the above-mentioned shortcomings, thus contributing to the spread of indoor vertical gardening. Our approach is suitable for both interior and exterior vertical gardens and for cultivation of both ornamental and edible plants. It can be matched with various architectural and interior design elements and can also easily be integrated with the technical systems of the building, as well as with the "smart house" concept. The current study provides a general summary of the results of the laboratory study conducted at Estonian University of Life Sciences [2,15] and the baseline study conducted at the University of Tartu. Vertical landscape planting systems consist of three major parts: a skeleton or frame holding the system, planting substrate and a watering system. Elaboration of long-span supporting structures from timber allows the prefabrication of details that can easily be mounted without using any fasteners and filled with ready-made concrete composite substrate blocks provided with all nutrients needed by plants. Such solutions allow reducing the manual work significantly as well as the overall costs. The demo setup was erected in the hall of Institute of Agriculture and Environmental Sciences, Estonian University of Life Sciences [16].

## 2. Materials and Methods

### 2.1. Timber Structures

Condensation on metal parts and deformations caused by uncontrolled environments are common problems with structures used in vertical gardening. These problems can be circumvented by using prefabricated wooden structures made of solid timber [17]. The details, which will be manufactured in the factory (off-site) using CNC processing at a controlled temperature and humidity corresponding to indoor conditions, can be easily assembled on the construction site. Wooden parts exposed to variations in ambient conditions will have different moisture contents (MCs) after assembly, and carpenter joints will "lock" due to different expansions and contractions of the different species of solid wood or different types of composited wood, respectively.

In the case of wet conditions, wooden construction elements, even if treated, cannot guarantee retaining the designed shape. To prevent plastic deformations because of different MC, curved, double-curved or even twisted compound or glulam (GL) beams or posts can be used to act as leaf springs. Some details can be assembled pre-stressed to have a preventive effect to keep the shape under the loads, taking into consideration and not overcoming the limitation of the creep values (max 3% in softwood, as known) [18].

Experience acquired during the assessment of historical buildings and antique woodworks has proven the durability and reliability of using old carpentry joints without the application of most metallic fasteners. Using contemporary CNC technologies enables developing similar long-lasting carpentry joints with even better designs. It is important that load-bearing constructions are made of coniferous softwood and the pins of deciduous wood (hardwood). To handle the expansion and shrinkage of timber details, the pins used need to have different line expansion coefficients to maintain a constant tolerance in carpentry joints and avoid ineligible extra inside forces [18].

These skills were well known and were at high levels in medieval times but are rarely used nowadays, even though the carpentry joints themselves are well known. Precision carpentry joints manufactured on 5-axis CNC workstations, combined with global knowledge from all existing databases of contemporary locally classified carpentry joints from

many countries (Japan, Norway, Russia, etc.), provide us with structural capabilities at a qualitatively new level.

Knowledge from the past and new information acquired through structural tests with wood were transferred to a prototype of a pyramidal shelf used in vertical gardening based on novel composite blocks (Figure 1).

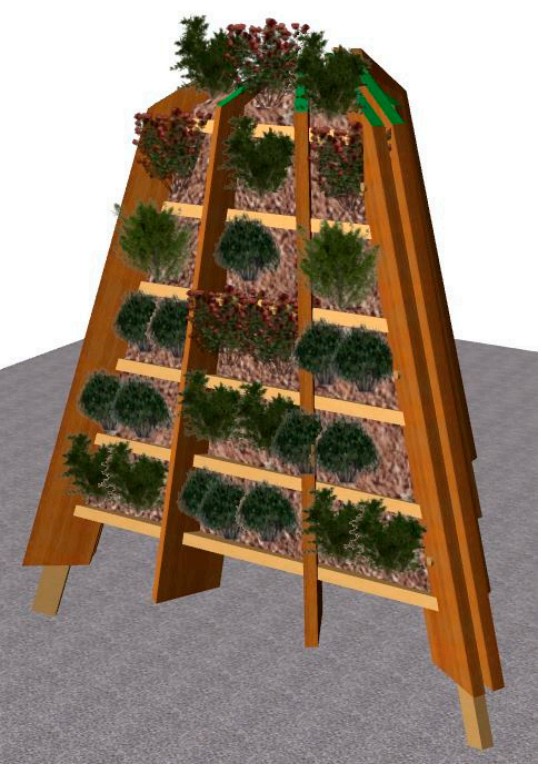

**Figure 1.** Prototype of vertical gardening pyramidal frame made of timber (joined with carpentry joints only).

### 2.2. Concrete Composite Substrates

The concrete composite used for the manufacturing of substrate blocks was described in our previous work [19]. Using this kind of material facilitates firmly manufacturing the details for vertical greenery, as the blocks of composite with the necessary shape and measures can be prefabricated and even furnished with seedlings. The manual work in situ incorporates just filling the prefabricated wooden structure with prefabricated blocks and equipping this with a watering system.

This kind of composite was invented at the University of Tartu and further developed at the Estonian University of Life Sciences (Tartu, Estonia) for 3D printing of buildings [19]. The composite has excellent thermal and building properties. Similar composites are also used for urban landscaping and gardening. The main components of composites for seeding or planting are as follows:

- Different types of peat or other organic fillers;
- Mixture of different kukersite oil shale ashes; fly ash from pulverized firing, desulfurizing ash with high $CaSO_4$ content and ash from incineration of oil shale in circulating fluidized bed process;
- Silica fume (nano-$SiO_2$), unneeded by-product of metallurgy (waste), mainly of steel production [20];
- Sulfur, for fast lowering of the pH of the composite;
- Various nutrients for plants: mineral and micronutrients;
- Humates to buffer the environment.

Patent application has been filed to the technology described by its authors. The patent "Concrete composite with enhanced ability of nutrients and water binding for plant growth and method of using the same" is pending.

## 3. Results and Discussion

### 3.1. Timber Structures

The use of timber structures in landscaping and gardening has a long history [21]. However, expansion because of moisture and shrinking because of drying and decay are still everyday problems of timber structures [22]. The conflict of timber vs. soil caused by the direct contact of water and aggressive minerals with timber and metallic parts still exists [23].

The decay of wood because of moisture and freshet damages take many timber structures out of use every year all over the world. Lightweight, renewable building materials, such as wood, and reusing old technologies combined with the possibilities of the Computer Age give us new opportunities to design and construct timber structures better for long life and different purposes, such as horizontal and vertical gardening.

Using treated timber in the abovementioned structures is common, but the traditional skills of how to use the orthotropic properties of wood in the best way and with the correct geometry of structures to avoid any kind of damage have decreased. Due to the life span of timber structures, those used in landscape architecture and gardening do not last long enough.

All the details were prefabricated using CNC processing at controlled temperature and humidity corresponding to indoor conditions and were assembled using traditional carpentry joints. Some joints are presented in Figure 2, and their modular assembly with carpentry joints is shown in Figure 3.

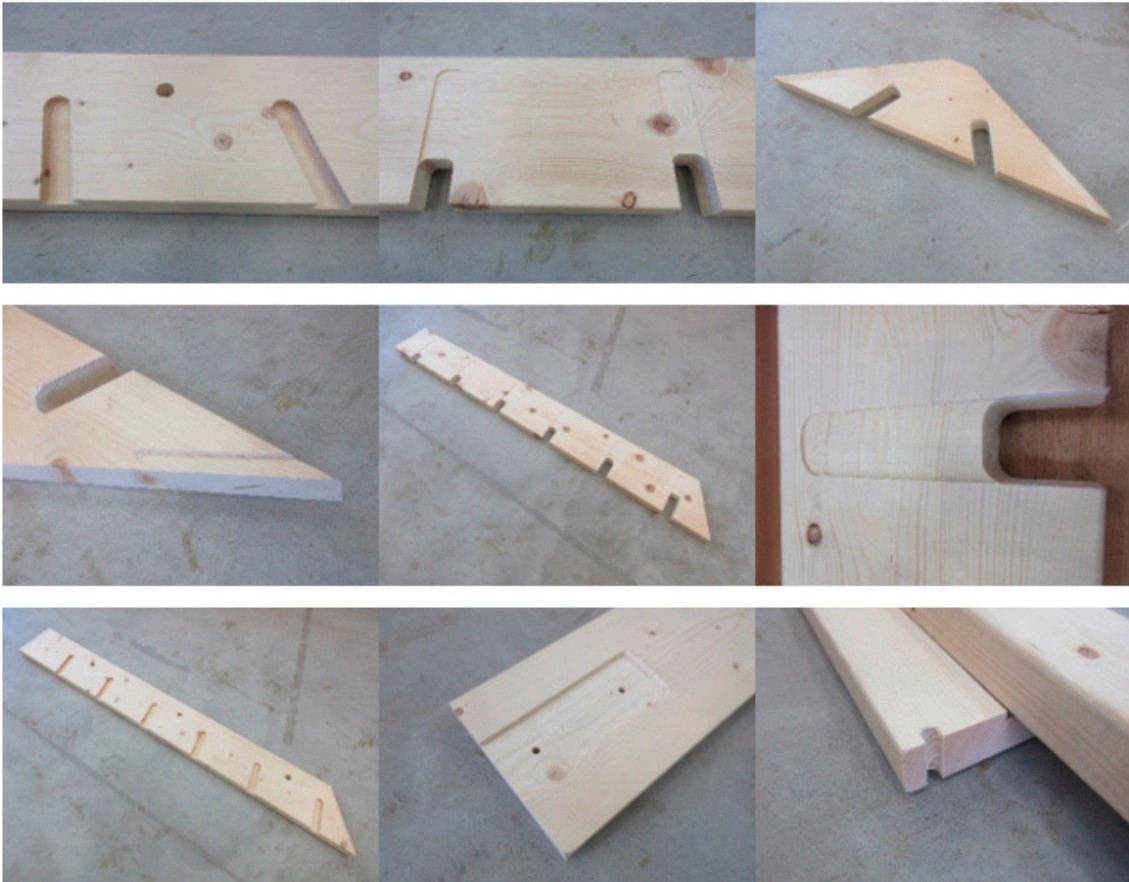

**Figure 2.** Some details and carpentry joints of the pyramidal frame structure.

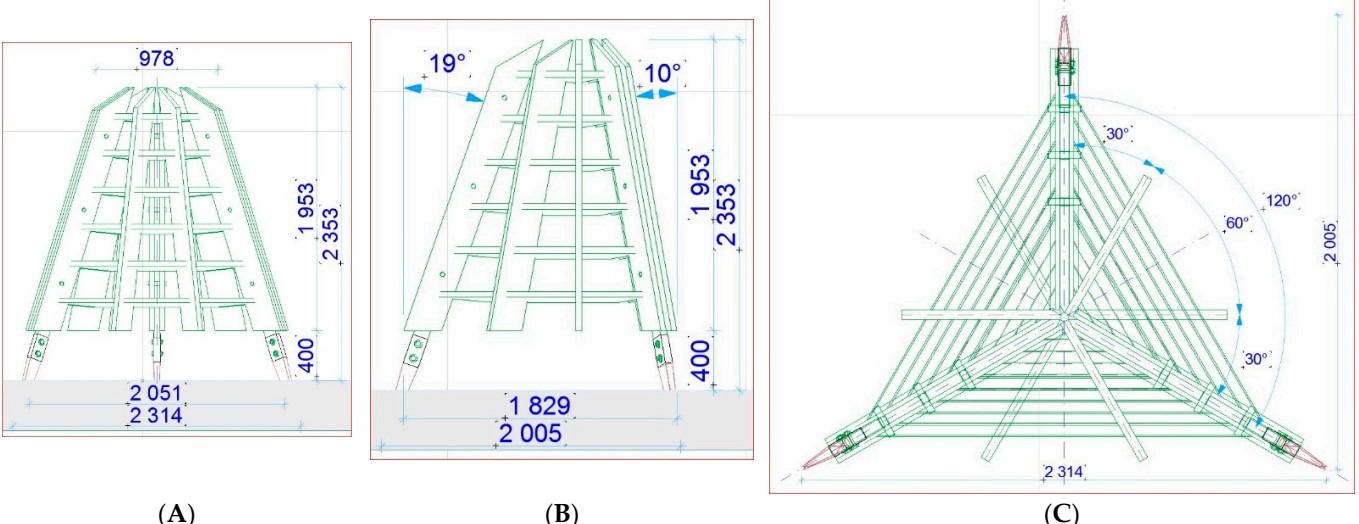

**Figure 3.** Modular assembly of pyramidal frame structure (**A–C**). (**A**) Front view; (**B**) side view; (**C**) top view.

### 3.2. Concrete Composite Substrates

In many cases, replacing the soil substrate with a self-sustaining composite is very useful. First, it avoids the use of plastic containers, pots, etc., needing recycling and the use of fossil raw materials. Second, the use of semi-hard composites decreases the run-off of the substrate into the watering system, which is extremely important for closed gardening systems.

Usually, this kind of composite includes an inorganic base (sand) glued together with a polymer binder, often super-adsorbent polymers (SAPs) [24,25], polyurethane [26] or other resin. The main shortcoming of these materials is the presence of plastic material inside the substrate; therefore, they are not fully biodegradable, and using them nevertheless produces a huge amount of plastic waste.

Our workgroup solved the problem by using the buffering properties of humates. This allows the retention of the concentration of soluble phosphates and maintenance of the functionality of the root system under the high pH level.

As the eco-footprint of cement production is extremely high, accounting for approximately 5% of atmospheric emissions of $CO_2$ in the world [27], Portland cement in some cases can be replaced with a mixture of ashes, including oil shale ash and pozzolanic additive (silica fume, soluble silicates, etc.) [19,28], using industrial waste as a raw material. Peat milling residues (highly decomposed peat), milled straw, sawdust and other inexpensive organic materials can be used as fillers.

Using cement-based concrete composites is a challenging way to avoid the use of polymers, but the main problem is bonding phosphorus into an insoluble apatite and therefore its unavailability for plants. The high pH of concrete is inconvenient for many plant species [29]. One solution is the use of porous concrete with a high void content as just a root holder media for the hydroponic growth of plants [30].

The fast hardening of blocks requires pH values as high as 13, but peat is acidic due to its high humic acid content. They also act as inhibitors for the pozzolanic hardening process. High pH allows the oxidation of humic acids by oxygen from air into oxalate, which precipitates as insoluble calcium compounds. The resulting decrease in the inhibition caused by humic matter allows the formation of insoluble alkaline feldspars, causing the pH to drop. When the pH drops below 12.4, $Ca(OH)_2$ is soluble in the pore water, and $Ca^{2+}$ ions migrate into the aqueous phase. Pozzolanic reactions and calcium oxalate precipitation can now take place. Eventually, the unconsumed $Ca(OH)_2$ absorbs $CO_2$ from the ambient air, forming $CaCO_3$ [19].

At the same time, a high pH is not suitable for plants; most of them are conformed to pH values of approximately 7 (for some species up to 8) [31]. A very complicated chain of chemical reactions was determined to assure fast and reliable hardening and, at the same time, fast lowering of the composite pH (Figure 4) to guarantee the best growing environment for most plant species.

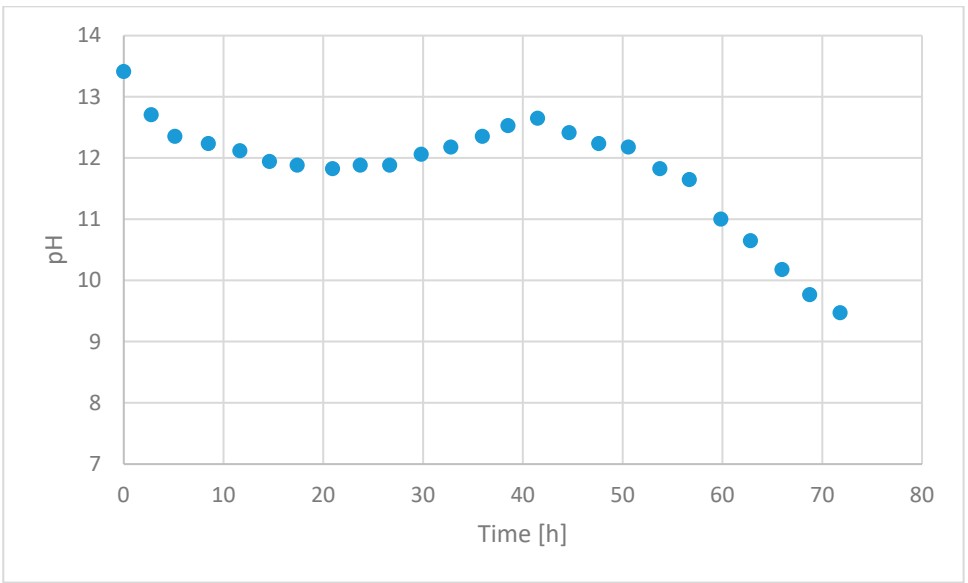

**Figure 4.** pH changes of concrete composite (time indicated in hours).

In substrates with high calcium content, the growth of plants is aggravated; to the best of our knowledge, no plant cultivation in substrates with such mineralization has been attempted to date. On thin (<20 cm) limestone, soils with an order of magnitude lower Ca content than our composite developed species-rich plant communities (*alvar* in Estonia) which are adapted to the environment plant biome [32]. Our goal, however, was to test as many different plant species as possible, which usually do not adapt to such conditions.

One of the main problems with such an environment is the availability of fertilizers. Potassium and nitrogen do not react directly with calcium ions, and various methods can be used to ensure the optimal concentration of these elements. However, there are major problems in making phosphorus available.

Both potassium and nitrogen as plant nutrients are essentially renewable over time—from bedrock and air nitrogen, respectively. Conversely, phosphorus is not very widespread in the Earth's crust and is very local; its migration is mainly mechanical (stool, dead organisms, etc.) [33]. Phosphorus occurs in soils as inorganic and organic forms, both of which can be absorbed by plants under certain conditions. Organic phosphorus from decomposed organisms can fall into different groups, usually dominated by phospholipids, nucleic acids, inositol phosphates and phosphoproteins, and inositol phosphates (phytate) [34].

Inorganic P in acidic soils consists mainly of compounds with iron and alumina; alkaline soils are dominated by different calcium phosphates (apatite) [35]. When water-soluble phosphorus fertilizers are applied to soils with high calcium content, the phosphorus will mostly be precipitated as dicalcium phosphate or octocalcium phosphate. The availability of phosphorus fertilizers added to soil will rapidly diminish due to its binding to calcium, iron and aluminum ions. It has been shown that various organic materials added to soil reduce the potency of P adsorption and increase its availability [35]. Likewise, the amount of phosphorus from organic fertilizers remains stable for a long time, while phosphorus is rapidly fixed from inorganic β-fertilizers [36]. Because of such strong interactions with the soil, the concentration of phosphates in calcareous soils is often very low [37]. To address such adverse conditions, plants have developed various mechanisms that influence the chemical conditions at the root–soil interface and promote the dissolution of soil P in the

rhizosphere [38]. To obtain inorganic P, the roots are able to release protons and organic anions, including oxalic, succinic, malic, tartaric and citric acids.

The addition of fulvic and humic acids improves the solubility of bound phosphorus, forming phosphate metal anion resin complexes that will be taken up by plant roots [39].

Different types of composite substrates have passed laboratory and field tests and showed good properties for plants growing in extreme conditions on facades and flat, inclined or curved roofs. We performed numerous tests with composite blocks; some specimens with different plants have passed a one-year open-air cycle (September 2018–September 2019). The material itself is hygroscopic, so it can absorb moisture/condensate during the night and allocate it inside the composite equally. If the environment becomes too wet, the outer surface expands until the humidity saturates and will prevent the covering from leaving too much water. Such kinds of physical processes make the conditions inside the composite stable for the roots of plants and avoid damage to them due to usual soil fissures.

Based on the results of both laboratory and field tests of 2 years thus far, the composite blocks have not been broken down due to watering and water splash caused by heavy rain. The composites developed by us bind nutrients and water for a longer time than conventional substrates due to the special properties of humates.

The nutrients are efficiently prevented from leaching out from the blocks. This has been proved by tests with various species of herbs, flowers, vegetables, potted plants, and even orchards. Most of the plants germinated and rooted well from sprouts and seedlings, being healthier than plants grown on conventional substrates with only dipping (see Figure 5).

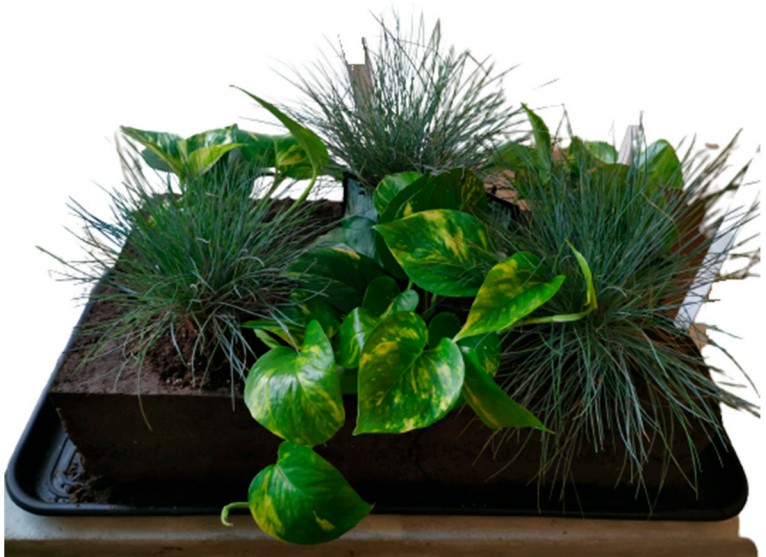

**Figure 5.** Composite block with plants.

### 4. Conclusions

The integrated and automated technology of CNC for wooden structures and pre-fabricated substrate blocks of concrete composite substrates allows us in the long run to significantly reduce the volume of manual work in landscaping and gardening and to use a large free-dimensional facade and/or roof as greenery surfaces. Composites do not contain any metal or plastic details or any other synthetic materials; thus, they are not harmful to the environment. Waste materials, such as wood and peat waste as well as other bioorganic materials and various ashes from the incineration of kukersite oil shale or biofuels, can be utilized directly, so the final product has a very low carbon footprint.

Our experiments showed that despite the high calcareous content of this type of composite, most plant species can successfully grow on them at least as well as on usual soil substrates or even better; just a few species requiring highly acidic soils cannot thrive. The balancing of different nutrients and their precisely timed release, correlating with

different phases of the plant's growth cycle, prolongs the decorative lifetime of some plant groups.

**Author Contributions:** Conceptualization, J.L., T.T. and M.S.; methodology, T.T., O.E., M.S. and J.L.; software, T.T.; validation, all authors; investigation, J.L., E.R. and T.T.; resources, all authors.; data curation, J.L., E.R. and T.T.; writing—original draft preparation, T.T., M.S., E.R. and J.L.; writing—review and editing, all authors; visualization, all authors.; supervision, T.T., M.S. and J.L.; project administration, T.T. and M.S.; funding acquisition, T.T. All authors have read and agreed to the published version of the manuscript.

**Funding:** This research was funded by project KIK 15392 by Estonian Environmental Investment Centre.

**Institutional Review Board Statement:** Not applicable.

**Informed Consent Statement:** Not applicable.

**Data Availability Statement:** Not applicable.

**Acknowledgments:** Merilin Noormets, Hille Lass, Maarja Kruus, Kadi Tõnurist, Iti-Miina Ulmas are acknowledged for their assistance in the experiments involving plants. Sander Saar, Märt Pilliroog, Karl-Martin Teppand, Sten Oliver Haugas and Markkos Mandel are acknowledged for their contribution to CAD-projecting, NC-programming and operating CNC-workstation Casadei-Busellato Jet optima T5 XL in producing, mounting and testing timber structures.

**Conflicts of Interest:** The authors declare no conflict of interest.

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
