# Peer review of "Timber Structures and Prefabricated Concrete Composite Blocks as a Novel Development in Vertical Gardening"

_sustainability, doi:10.3390/su142114518_

Round 1

Reviewer 1 Report

This manuscript presents "Timber composite structures and prefabricated concrete composite blocks as a novel development in vertical gardening." The topic is interesting and fits well with the scope of the journal. There are, however, many inconsistencies within the manuscript that must be clarified prior to acceptance for publication can be recommended. Please read the “instructions for authors” of the Sustainability journal, and I recommend carefully reading the manuscript and revising it based on author guidelines (e.g., citations).

Major comments:
1. English should be checked.
2. Missing citation in the introduction section - especially page 2.
3. What does the novelty of this study? Please clearly mention in the introduction section, as well as the research gap.
4. Research methods were not clearly mentioned in the manuscript, and it will be acceptable to explain them more clearly (step by step) in the methodology section.
5. Authors have mentioned many unclear statements in the results and discussion section without proper explanation in the methodology section
(e.g., page 9, "Based on the results of both laboratory and field tests (2 years thus far)"). Laboratory and field test – more clarification is needed.
Also, please carefully read the results and discussion section and provide more details where necessary.
6. Word "watering system" should be clearly explained in the manuscript.
7. Reference errors (e.g., reference no. 4, 6). Please read the author guidelines of the journal.

Specific comments:
1. Use one word - Figure or Fig. - read the manuscript and revise where necessary.
2. Figure captions should be explained more clearly.
3. Page 7. Some words - Check the font size
4. Figure caption should be mentioned under the figure.
5. Figure 3. A clear explanation is needed. Missing information. X-axis and y-axis labels
6. Page 9. Photograph – please include the caption (e.g., Photograph x: …..), and it should mention in the main text before the photograph.

Author Response

Dear Reviewer 1. 

Hereby, please find our changes in to the manuscript as indicated in your review. 

Replies to “major comments”:

1: The manuscript has undergone language editing and the English has been carefully checked.

2: The text has been checked for erroneous references and incorrect references found by the reviewer have been fixed. The text has also been checked for missing references and new references have been added.

3: To emphasize the innovative nature of this study and highlight the gaps in previous work that this study intends to bridge, the following paragraphs have been added to the introduction:

However, the existing engineering solutions described in the literature typically rely on the use of substantial amounts of plastic and metal parts and details.”

“This paper proposes a new conceptual approach to modern methods of designing and maintaining vertical gardens that addresses all the above-mentioned shortcomings, thus contributing to the spread of indoor vertical gardening. Our approach is suitable for both interior and exterior vertical gardens and for cultivation of both ornamental and edible plants. It can be matched with various architectural and interior design elements and can also easily be integrated with the technical systems of the building, as well as with the "smart house" concept.”

4: The description of the research methods has been improved also, we added some clarification material  on figure 3.

5: The results and discussion section has been rewritten in order to provide more detailed analysis of the laboratory work.

6: The description of the watering system has been rewritten to provide more detailed information.

  1. The erroneous references have been fixed.

Replies to “specific comments”:

1: The word “Fig” was used for figures throughout the text. The body text and figure captions were revised accordingly.

2: Figure captions were changed, please see the newed captions.

3: The font size has been unified throughout the text.

4: All figure captions were placed under the figures.

5: Explanation was added to the text about Figure 3. The Figure was revised and new information added.

6: Figure caption was added to the photo in page 9 and reference to this figure was added to the main text.

Kindly yours, authors

Reviewer 2 Report

The paper is relevant for the journal. However, The case study should have less weight in the title and in the abstract, and the innovative methodology should be emphasized.The method in itself is good and well described.

Conclusion and discussion Needs to be expanded,Some additional comparative literature is needed to clarify the baseline for research innovation.

Author Response

Dear Reviewer 2

New references were added to the paper to expand and clarify the baseline for research innovation. The innovative methodology was emphasized.  Changes were made in the conclusions and results and discussion sections.

Kindly yours authors

Round 2

Reviewer 1 Report

With sound justifications, the authors have mostly revised the paper. I believe the manuscript is suitable for publication after minor revisions. 

However, please read the "Instructions for Authors" for citation style ("References: In the text, reference numbers should be placed in square brackets [ ], and placed before the punctuation; for example [1], [1–3] or [1,3]. For embedded citations in the text with pagination, use both parentheses and brackets to indicate the reference number and page numbers; for example, [5] (p. 10). or [6] (pp. 101–105").

Figure 4. It is acceptable to begin the y-axis at 5 or 6.

Author Response

Dear reviewer, 

please find from attachement corrected version of our manuscript. 

The corrections are made as following 

1. We corrected the citation style and removed doubled citation source. 

However, please read the "Instructions for Authors" for citation style ("References: In the text, reference numbers should be placed in square brackets [ ], and placed before the punctuation; for example [1], [1–3] or [1,3]. For embedded citations in the text with pagination, use both parentheses and brackets to indicate the reference number and page numbers; for example, [5] (p. 10). or [6] (pp. 101–105").

2. We corrected the figure and changed the y-axis starting point. 

Figure 4. It is acceptable to begin the y-axis at 5 or 6.

Please, let us know if the corrections are sufficient. 

Kindly yours, corresponding author
